# Oxalactam A, a Novel Macrolactam with Potent Anti-*Rhizoctonia solani* Activity from the Endophytic Fungus *Penicillium oxalicum*

**DOI:** 10.3390/molecules27248811

**Published:** 2022-12-12

**Authors:** Ruizhen Zhang, Yingrun Ma, Ming-Ming Xu, Xinyi Wei, Cheng-Bin Yang, Fei Zeng, Jin-Ao Duan, Chun-Tao Che, Junfei Zhou, Ming Zhao

**Affiliations:** 1National and Local Collaborative Engineering Center of Chinese Medicinal Resources Industrialization and Formulae Innovative Medicine, Jiangsu Collaborative Innovation Center of Chinese Medicinal Resources Industrialization, School of Pharmacy, Nanjing University of Chinese Medicine, Nanjing 210023, China; 2Department of Pharmaceutical Sciences, College of Pharmacy, The University of Illinois at Chicago, Chicago, IL 60612, USA

**Keywords:** oxalactam A, macrolactam, anti-*Rhizoctonia solani*, *Penicillium oxalicum*, molecular docking, molecular dynamics

## Abstract

A novel macrolactam named oxalactam A (**1**), three known dipeptides (**2**–**4**) as well as other known alkaloids (**5**–**7**) were obtained from the endophytic fungus *Penicillium oxalicum*, which was derived from the tuber of *Icacina trichantha* (Icacinaceae). All chemical structures were established based on spectroscopic data, chemical methods, ECD calculations, and ^13^C-DP4+ analysis. Among them, oxalactam A (**1**) is a 16-membered polyenic macrolactam bearing a new skeleton of 2,9-dimethyl-azacyclohexadecane core and exhibited potent anti-*Rhizoctonia solani* activity with a MIC value of 10 μg/mL in vitro. The plausible biosynthetic pathway of **1** was also proposed via the alanyl protecting mechanism. Notably, three dipeptides (**2**–**4**) were first identified from the endophytic fungus *P. oxalicum* and the NMR data of cyclo(*L*-Trp-*L*-Glu) (**2**) was reported for the first time. In addition, the binding interactions between compound **1** and the sterol 14*α*-demethylase enzyme (CYP51) were studied by molecular docking and dynamics technologies, and the results revealed that the 16-membered polyenic macrolactam could be a promising CYP51 inhibitor to develop as a new anti-*Rhizoctonia solani* fungicide.

## 1. Introduction

Rice sheath blight (RSB), also called “snakeskin disease”, “mosaic footstalk”, and “rotten footstalk”, is one of the most devastating rice diseases caused by the necrotrophic pathogen *Rhizoctonia solani* [1,2]. The RSB incidence has witnessed a sharp increase owing to the unrestricted nitrogen fertilizers and semi-dwarf high-yield cultivars in recent decades [3]. In China, RSB was already the second most serious disease in rice, which could lead to a yield loss of 10~30% with 15~20 million hm^2^ every year [4,5], even up to 50% in the Yangtze river valley in epidemic years [6,7]. Currently, chemical fungicides are major approaches to preventing RSB and related diseases with increased human health risks, environmental pollution, and resistant phytopathogens [8]. Therefore, natural anti-*R. solani* agents, such as strobilurins or QoI from *Strobilurus tenacellus* (wild mushroom) [8], have engaged worldwide attention owing to their environmental friendliness, high selectivity, and new mechanism of action.

Recently, macrolactam scaffolds are usually regarded as precursor components with high efficacy including antibiotics, antifungals, anticancer as well as immunosuppressants [9]. A typical macrocyclic glycopeptide antibiotic, vancomycin, has been applied to treat Gram-positive bacterial infections including methicillin-resistant *Staphylococcus aureus* and penicillin-resistant *Streptococcus pneumonia* in the clinic for a long time [9]. Previous investigations indicated that 59% of approved small molecules bearing a macrolactam core are natural products or related derivatives [10]. Therefore, macrolactam components might be a promising lead scaffold for pharmaceutical purposes [11]. However, there is no report on their anti-*R. solani* activity in agricultural applications.

*Icacina trichantha* Oliv. (Icacinaceae), distributed in West and Central Africa [12], is an excellent plant resource that has the concomitant function of both medicine and foodstuff to relieve constipation, food poisoning, and malaria [13]. Due to the tropical rainforest climate near the equator in Nigeria, *I. trichantha* might be a potential plant source to seek various bioactive endophytes producing interesting secondary metabolites. Until now, no report on the endophytic fungus derived from *Icacina* species and related bioactivity has been documented. Previously, our group found that *Penicillium oxalicum* from the tuber of *I*. *trichantha* could significantly reduce the growth of *R. solani* strain with a 96.83% of inhibition rate (*Alternaria alternata*, 83.33%; *Fusarium solani*, 50.67%; *Colletotrichum gloeosporioides*, 76.04%) by the plate confrontation test (Appendix A). Therefore, the further anti-*R. solani* activity-guided fractionation resulted in the isolation of one novel macrolactam (**1**), three dipeptides (**2**–**4**) as well as other alkaloids (**5**–**7**) (Figure 1). Notably, oxalactam A (**1**) represents the first example of a 16-membered polyenic macrolactam bearing a 2,9-dimethyl-azacyclohexadecane core, and this is the first report of the NMR data for cyclo(*L*-Trp-*L*-Glu) (**2**). Additionally, three dipeptides (**2**–**4**) were first identified from the endophytic fungus *P. oxalicum*. Among them, oxalactam A (**1**) exhibited potent anti-*R. solani* activity with a MIC value of 10 μg/mL by the mycelium growth rate method in vitro. Herein, the isolation, structural elucidation, anti-*R. solani* activity, molecular docking, and molecular dynamics simulation of these isolates were elaborated.

## 2. Results and Discussion

### 2.1. Isolation and Structural Identification

Oxalactam A (**1**), [*α*${]}_{D}^{20}$ + 2.00 (*c* 0.1, MeOH), was isolated as a white amorphous powder. Its molecular formula could be verified as C_23_H_37_NO_9_ in line with the HR-ESI-MS at *m/z* 518.2943 [M + CH_3_CH_2_OH + H]^+^ (calcd for C_25_H_44_NO_10_, 518.2965) and its ^13^C NMR data, indicating six degrees of unsaturation. The presence of double bonds in **1** could be assigned based on a maximum absorption at 207 nm in the UV spectrum. The ^1^H NMR spectrum (Table 1) showed resonances attributed to five olefinic protons at *δ*_H_ (5.84, 5.72, 5.49, 5.45, 5.15), two oxygenated methines at *δ*_H_ (4.44, 4.14), and one glucopyranosyl group in **1**. Additionally, the combination of ^13^C NMR, DEPT135, and HSQC data suggested the presence of a methyl at *δ*_C_ 16.3, a carbonyl at *δ*_C_ 175.6, three double bonds at *δ*_C_ (136.9, 134.8, 134.7, 131.2, 129.2, 125.0), and a glucopyranosyl segment at *δ*_C_ (104.9, 78.1, 78.0, 75.1, 71.7, 62.8) in **1**. Three double bonds, a carbonyl, and a glucopyranosyl account for five degrees of unsaturation, the remaining one demonstrated the presence of a monocycle system in the aglycone of **1**.

The ^1^H−^1^H COSY and HSQC data (Figure 2) suggested the presence of three spin systems in **1**, (a) H-3/H-4/H-5/H_2_-6/H_2_-7/H_2_-8, (b) H-10/H_2_-11/H_2_-12/H-13/H-14/H-15/H-16/H_2_-17, and (c) H-1′/H-2′/H-3′/H-4′/H-5′/H_2_-6′. Additionally, HMBC correlations (Figure 2) from one methyl singlet H_3_-18 to C-8/C-9/C-10 suggested the direct connections of C-8, C-10, and C-18 toward C-9. The connection between C-2 and C-3 was verified by the key HMBC correlation from H-3 to C-2. The HMBC correlation from the H-1′ to C-17 established the connections of one glucopyranosyl group to the oxygenated primary carbon C-17. Furthermore, H-16 (*δ*_H_ 3.97)/C-16 (*δ*_C_ 54.7) were different compared to a typical oxygenated secondary carbon, thus, a nitrogen bridge exists between C-2 and C-16 according to the molecular formula and resulting degrees of saturation of **1**. Notably, all three *trans* double bonds Δ^4,5^, Δ^13,14^, and Δ^9,10^ could be established by the coupling constants of *J*_H-4/H-5_ = 16.0 Hz, *J*_H-13/H-14_ = 15.1 Hz, and the key NOESY correlation between H-8 and H-10. Consequently, the planar structure of oxalactam A (**1**) was constructed as (4*E*,9*E*,13*E*)-3,15-dihydroxy-16-(glucopyranosyl-*O*-methyl)-9-methyl-azacyclo-hexadeca-4,9,13-trien-2-one.

However, due to the flexibility of the 16-membered ring, the relative configuration of oxalactam A (**1**) could not only be constructed from the NOESY data and require a special assignment. The stereochemistry of C-3 and C-15 were confirmed according to the DP4+ probability for theoretical ^13^C NMR shifts of four possible isomers (3*R**,15*S**,16*S**)–**1**, (3*R**,15*R**,16*S**)–**1**, (3*S**,15*S**, 16*S**)–**1**, and (3*S**,15*R**,16*S**)–**1**. Finally, (3*R**,15*S**,16*S**) could be determined as the relative configuration of **1** with a 97.08% DP4+ probability (Figure 3) [14].

To confirm the assigned structure of **1**, (3*R**,15*S**,16*S**)–**1**/(3*R**,15*R**,16*S**)–**1**/(3*S**,15*S**, 16*S**)–**1**/(3*S**,15*R**,16*S**)–**1** were further analyzed by the artificial neural networks (ANNs) method [15]. The NMR data of four possible isomers were recalculated at the OPT/HF/3-21G and GIAO/PCM/mPW1PW91/6-31G(d,p) levels in methanol, the ANNs calculation results classified four possible isomers of **1** into category 1 (correct) based on the analysis of 18 parameters (Appendix A), and the ratio of category 1 (correct) to category 2 (incorrect) was calculated as 0.8573:0.1198 for (3*R**,15*S**,16*S**)–**1** [0.3963:0.5382 for (3*R**,15*R**,16*S**)–**1**, 0.4834:0.4027 for (3*S**,15*S**,16*S**)–**1**, and 0.6683:0.2713 for (3*S**,15*R**,16*S**)–**1**], confirming the relative configuration of **1** as (3*R**,15*S**,16*S**)–**1**.

Oxalactam A (**1**) could be proposed as a 3-aminobutyrate unit in the starter position based on the *β*-amino acid incorporation pathway in polyketide macrolactam biosynthesis reported previously [16]. Therefore, the absolute configuration of C-16 in **1** could be assigned as *S*. Moreover, the coupling constant of H-1′ (*J* = 7.8 Hz) assigned the *β*-glucopyranosyl linkage in **1**, and the *D*-glucose was proved by the comparison of the HPLC retention times of the monosaccharide derivative of hydrolysate of **1** (*t*_R_ = 21.924 min) with those of the standards *D*-glucose (*t*_R_ = 22.179 min) and *L*-glucose (*t*_R_ = 19.962 min) (Appendix A). Further, the absolute configuration of **1** was verified by the electronic circular dichroism (ECD) method with time-dependent density functional theory [17]. The positive Cotton effect (222.4 nm) of **1** was consistent with that (224.2 nm) in the theoretically calculated ECD spectrum of (3*R*,15*S*,16*S*)–**1** (Figure 4). Thus, compound **1** was constructed as depicted and given a trivial name oxalactam A.

Oxalactam A (**1**) was the first example of a 16-membered polyenic macrolactam bearing a 2,9-dimethyl-azacyclohexadecane scaffold, and its biosynthetic pathway was also similar to those of vicenistatin from *Streptomyces halstedii* [18,19]. As shown in Figure 5, oxalactam A (**1**) biosynthesis originated from *L*-glutamate, which could convert into 3-aminobutyrate by VinH/VinI [20]. VinN, a adenylation enzyme, could recognize 3-aminobutyrate and transfer it to 3-aminobutyrate-VinL, which was aminoacylated with *L*-alanine under the catalysis of VinM to generate a dipeptidyl-VinL. The dipeptidyl group could be tied to the ACP domain of the modular PKS VinP1 by VinK. Then, modular PKSs VinP1–P4 elongated the polyketide chain with a terminal alanyl group, which could be removed by VinJ before intermediate **IV** generation by VinP4 thioesterase domain. Finally, the hydroxylase and glycosyl-transferase could successfully catalyze **IV** to yield oxalactam A (**1**). This alanyl protecting mechanism was also detected in the biosynthetic pathways of some other antibiotics including butirosin [21,22] and desertomycin [23].

Besides, three known dipeptides (**2**–**4**) and other alkaloids (**5**–**7**) were identified as cyclo(*L*-Trp-*L*-Glu) (**2**) [24], cyclo(*L*-Pro*-L*-Phe) (**3**) [25,26], cyclo(*L*-Pro-*L*-Tyr) (**4**) [27], penipanoid A (**5**) [28], (*Z*)-N-(4-Hydroxystyryl)-carboxamide (**6**) [28], and (*E*)-N-(4-Hydroxystyryl)-carboxamide (**7**) [28], by the specific rotation, HR-ESI-MS, and NMR data analyses, as well as the comparison with the literature data. Significantly, three dipeptides (**2**–**4**) were first identified from the fermentation liquid of *P. oxalicum*, and the NMR data of cyclo(*L*-Trp-*L*-Glu) (**2**) was reported for the first time. These isolated alkaloids in our study enriched the chemical diversity of secondary metabolites from *Penicillium* genus.

### 2.2. Anti-Rhizoctonia solani Activity

The previous investigation indicated that *P. oxalicum* is a promising fungal agent to prevent plenty of plant diseases [29], including *Phytophthora* root rot of azalea [30], *Pythium* seed rot [31], and tomato *Fusarium* and *Verticillium* wilts [32]. These applications have aroused our interest to search for active secondary metabolites from the endophytic fungus *P. oxalicum* as anti-*R. solani* drugs to relieve rice sheath blight. Consequently, all isolates **1**–**7** were evaluated for their anti-*R. solani* activity using the mycelium growth rate method in vitro. Among these alkaloids, compound **1** was the most active component with a 29.30% of inhibition rate at the concentration of 100 µM (MIC = 10 μg/mL), whereas compounds **2**–**7** did not show potent anti-*R. solani* activity under the same condition (Table 2, Appendix A). Up to now, this is the first 16-membered polyenic macrolactam with anti-*R. solani* activity for agricultural tools.

### 2.3. Assessment of Binding Affinity of **1** and CYP51

The 14*α*-demethylase enzyme (CYP51, PDB ID: 3GW9) [34] is an essential component of the fungal cell membrane, which played an important role in the fungal-specific ergosterol biosynthesis pathway. Azole antifungals (hexaconazole for example) could compete with the CYP51 substrate by binding to the heme iron in the active site in a ligand–binding pocket, leading to ergosterol depletion, membrane fluidity reduction, and lipid layer destruction [35]. To analyze the binding mode of compound **1** to CYP51 enzyme, molecular docking and dynamics studies have been performed using Discovery Studio 2019. The result illustrated that 18-methyl of **1** showed favorable steric interactions with LUE356, MET460, and VAL461 residues (Figure 6), and MET460 might cause a misfolded fungus-specific loop that affects the binding efficiency of the cognate NADPH-cytochrome P450 reductase [36]. Moreover, the 1,2,4-triazole and 1,3-dichlorobenzene moieties of hexaconazole were observed to generate interactions with the ALA291, CYS422, and ILE423 (Figure 7), and the ALA291 substitution is vital for the microbial resistance to triazole drugs including voriconazole and fluconazole [37]. Notably, it showed that the higher negative binding energy value of **1** (−202.0196 kcal/mol) compared to that of hexaconazole (−105.4279 kcal/mol) indicated more favorable binding interactions between compound **1** and the CYP51 enzyme (Table 3) [33], which could explain the reason for the superiority of compound **1** over hexaconazole.

The steric interactions between CYP51 enzyme and oxalactam A (**1**)/hexaconazole were further estimated by molecular dynamics simulation according to RMSD (Root-mean-square deviation), RMSF (Root-mean-square fluctuation), and the total energy (Figure 8). Accordingly, RMSD values suggested that both complexes could reach an equilibrium state at 0.6–0.9 nm of the vibration amplitude within 200 ps. Additionally, the active and binding sites (THR295–LEU359/GLY414–VAL461) of CYP51–**1** have a slightly higher RMSF than those of the CYP51–hexaconazole, indicating the CYP51–hexaconazole complex possesses better thermodynamic stability. Notably, the CYP51–**1** complex showed lower total energy values than that of CYP51–hexaconazole with 55 kcal/mol per picosecond, which demonstrated that the binding affinity of CYP51–**1** was much stronger than that of CYP51–hexaconazole. These results indicated that compound **1** could directly bind to the high-affinity catalytic areas of the CYP51 enzyme and yield stable enzyme–ligand complexes in the saline condition.

## 3. Materials and Methods

### 3.1. General Experimental Procedures

A UV-2600 spectrophotometer (Shimadzu, Kyoto, Japan) was applied to measure the UV absorption spectra. An MCP 150 digital polarimeter (Anton Paar, Graz, Austria) was utilized to gain the optical rotations. A J-810 spectrometer (JASCO, Kyoto, Japan) could be exerted to acquire CD spectra in methanol at room temperature. The HR-ESI-MS data could be collected by a Waters SYNAPT G2-Si Q-TOF mass spectrometer (Waters, Milford, MA, USA). The NMR spectra were recorded on a Bruker AV-500 spectrometer (Bruker, Karlsruhe, Germany). The semi-preparative RP-HPLC included a dual-wavelength detector, a Shimadzu LC-20AR instrument, and an Ultimate XB-C_18_ column (10 × 250 mm, 5 μm, Welch).

### 3.2. Strain Material

The test endophytic fungus was obtained from the fresh tubers of the West African plant *I. trichantha*, the Orba village in Nsukka of the Enugu State, Nigeria, in June 2011. It was identified as *P. oxalicum* based on its morphological characteristics and 16S rRNA gene sequence by Shanghai Shenggong Bioengineering Co., Ltd. A voucher sample (ZRZ20210301) was stored in the Key Laboratory of prescriptions of Nanjing University of traditional Chinese medicine.

### 3.3. Fermentation and Isolation

The potato dextrose agar (PDA) was used to culture the *P. oxalicum* strain at room temperature for 72 h. Next, the PDA was cut into small pieces and then inoculated into a mass of sterilized Erlenmeyer flasks containing 10 g glucose, 4 g soluble starch, 2.5 g peptone, 1 g sodium chloride, 1 g calcium carbonate, 0.25 g magnesium sulfate, 1 g yeast extract, 0.25 g dipotassium hydrogen phosphate, and 500 mL distilled water. All flasks were incubated at room temperature for 72 h. The culture broths (82 L) and the smashed mycelium were extracted by ethanol three times, then combined and concentrated under reduced pressure to yield a crude extract, which was partitioned between ethyl acetate and water to obtain the EtOAc-soluble extract (42.58 g). This part was loaded on a 200~300 mesh silica gel eluted with CH_2_Cl_2_–MeOH (100:0 to 0:100, *v*/*v*) to furnish eight fractions (Fr.1~Fr.8).

Fr.6 (3.67 g) was separated by MPLC with an MCI column to yield seven subfractions Fr.6-1~Fr.6-7 (MeOH–H_2_O, 10:90 to 100:0, *v*/*v*). Compound **2** (12.71 mg) was purified from Fr.6-6 (80.60 mg) by a semi-preparative HPLC with an Ultimate XB-C_18_ column (MeOH–H_2_O, 45:55, *v*/*v*, 3 mL/min). Fr.6-8 (1.47 g) was treated with excess sodium carbonate, then loaded on a 200~300 mesh silica gel CC using CH_2_Cl_2_–MeOH (10:1, *v*/*v*) mixed solvent to yield compound **1** (140.59 mg). Fr.3 (4.58 g) was isolated as nine sub-fractions (Fr.3-1~Fr.3-9) by a Sephadex LH-20 column with pure methanol. A semi-preparative HPLC (MeOH–H_2_O, 30:70, *v*/*v*, 3 mL/min) was applied to purify compound **3** (3.02 mg) from Fr.3-3 (0.78 g). Compounds **6** (10.88 mg) and **7** (5.84 mg) were also purified from Fr.3-5 (0.39 g) by a semi-preparative HPLC (MeOH–H_2_O, 20:80, *v*/*v*, 3 mL/min). Fr.4 (11.18 g) was acquired by MPLC with an MCI column (MeOH–H_2_O, 5:95 to 100:0, *v*/*v*) to afford nine fractions Fr.4-1~Fr.4-9. Compounds **4** (101.00 mg) and **5** (3.50 mg) were finally purified from Fr.4-7 (0.58 g, MeOH–H_2_O, 17:83, *v*/*v*, 3 mL/min) and Fr.4-9 (0.21 g, MeOH–H_2_O, 35:65, *v*/*v*, 3 mL/min) by a semi-preparative HPLC, respectively.

Oxalactam A (**1**): A white amorphous powder; [*α*${]}_{D}^{20}$+2.00 (*c* 0.1, MeOH); UV (MeOH) *λ*_max_ (log *ε*) 207 (3.24) nm; CD (MeOH): *λ*_max_ (Δ*ε*) 222.0 (+1.30) nm; ^1^H NMR (CD_3_OD, 500 MHz) data, see Table 1; ^13^C NMR (CD_3_OD, 125 MHz), see Table 1; HR-ESI-MS *m/z* 518.2943 [M + CH_3_CH_2_OH + H]^+^ (calcd for C_25_H_44_NO_10_, 518.2965).

### 3.4. Enzymatic Hydrolysis

Oxalactam A (**1**) (2 mg) and *β*-cellulase (0.5 mg) were dissolved in an aqueous solution (2 mL) at 50 °C for 4 h. The reaction solution was extracted by EtOAc five times, 5 mL each. The EtOAc extract liquor was combined and evaporated, and analyzed by the TLC and HPLC methods. Finally, the aglycone of compound **1** was purified by a semipreparative HPLC with an Ultimate XB-C_18_ column.

### 3.5. Sugar Identification

The Enzymatic Hydrolysis mixtures were analyzed using HPLC with an Ultimate XB-C_18_ column (CH_3_CN/H_2_O/CH_3_COOH = 25:75:0.1, 0.8 mL/min). The sugar moiety of *D*-glucose in **1** (*t*_R_ = 21.924 min) was assigned by the comparison of the retention time of the monosaccharide derivative with those of *D*-glucose (*t*_R_ = 22.179 min) and *L*-glucose (*t*_R_ = 19.962 min) [38].

### 3.6. ECD and NMR Calculation Methods

HyperChem 8.0 program was carried out to search for the most stable stereoisomers of compound **1** with molecular mechanics MMFF94s. Gaussian 16 software was applied to optimize geometries to screen minimum-energy conformers at the OPT/B3LYP/6-31+G(d) (OPT/HF/3-21G for ANNs analysis) level. The optimized conformers were submitted to the ECD calculation program with the TD/B3LYP/6-311+G(d) level in methanol. The overall ECD curves were summed up by the Boltzmann distribution averaging of all conformers with SpecDis 1.71 software [39]. Additionally, the ^13^C NMR shifts calculation of all stable conformers of **1** was conducted with the GIAO/PCM/mPW1PW91/6-311G(d,p) (GIAO/PCM/mPW1PW91/6-31G(d,p) for ANNs analysis) level in methanol. The DP4+ probability and linear regression were obtained by analyzing overall theoretical NMR data to verify the relative configurations of C-3/C-15 in **1** [14].

### 3.7. The Plate Confrontation Test

The fungal blocks of *Rhizoctonia solani*, *Alternaria alternata*, *Fusarium solani*, and *Colletotrichum gloeosporioides* (each diameter 5 mm) were, respectively inoculated in the center of the PDA plate (each diameter 150 mm), and purified endophytic fungus *P. oxalicum* was inoculated in the surroundings. All plates were incubated at room temperature for 72 h. Each treatment was repeated three times and the sterile water was used as a blank control. The antagonistic effect was determined by the width of the inhibition belt when colonies fully covered the whole Petri dish in blank control.

### 3.8. Anti-Rhizoctonia solani Assays In Vitro

The sensitivity of *R. solani* to compounds **1**–**7** from the endophytic fungus *P. oxalicum* was measured by the mycelium growth rate method in vitro. This phytopathogenic fungus, *R. solani*, was provided by Beijing Baioubowei Biotechnology Co., Ltd.

A mycelial plug (1 cm in diameter) was cut from the beforehand PDA medium fully covered by *R. solani* colonies, which was placed with their mycelia-side down on the PDA medium in the center of each plate containing 100 µM of pure compounds **1**–**7**. The positive control is hexaconazole (100 μM), a commercial fungicide. An equal concentration of DMSO solution served as the solvent control. A thermostatic chamber was applied to inoculate the treatment plates, and the average diameter of each *R. solani* colony was recorded based on the previous literature [40]. The anti-*R. solani* activity was evaluated in terms of the inhibition rate, which was performed in triplicate for each treatment. The inhibition ratio could be acquired by the following formula: Inhibition (%) = (A − B)/A × 100% (A: average diameter of the control group, B: average diameter of the treatment group) [41].

### 3.9. Molecular Docking Study

The molecular docking simulation [42] of compound **1** and hexaconazole to the sterol 14α-demethylase enzyme (CYP51, PDB ID: 3GW9) was performed by the CDOCKER protocol of Discovery Studio 2019 (BIOVIA, San Diego, CaA). The CYP51 X-ray crystal structure was downloaded from RCSB Protein Data Bank (http://www.rcsb.org, accessed on 1 June 2022). All conformations of compound **1** were searched by ChemHyper 8.0 program with the density functional method (6-31G*). The CYP51 protein was protonated and deleted water at pH 7 using Prepare Protein tool under the CHARMm forcefield [43]. Additionally, the implicit solvent model was selected as the Generalized Born with Molecular Volume method. Additionally, the binding energy of CYP51–**1**/hexaconazole was calculated by the Spherical Cutoff mode. All other parameters were set as default.

### 3.10. Molecular Dynamics Simulation

Molecular dynamics simulation [42] was also performed for CYP51–**1**/hexaconazole based on molecular docking results using the Standard Dynamics Cascade protocol of Discovery Studio 2019 (BIOVIA, San Diego, CA, USA). This software brings together over 30 years of peer-reviewed research and world-class in silico techniques such as molecular mechanics, free energy calculations, and biotherapeutics developability and more into a common environment, and it could be available at the website: https://www.3ds.com/products-services/biovia/products/molecular-modeling-simulation/biovia-discovery-studio/ (accessed on 1 January 2022).

The CYP51–**1**/hexaconazole complex was in the aqueous environment with the Extended Simple Point Charge water model under the CHARMm force field [44]. The entire system was equilibrated under NVT (isothermal-isochoric) and NPT (isothermal-isobaric) ensembles to maintain the stabilized pressure [45]. All hydrogen bonds were constrained during equilibration using LINC algorithms [46]. Besides, the Particle Mesh Ewald module was applied for the long-range ionic interaction [47]. Finally, the entire trajectories were saved for analysis at a frequency of 1 ps.

## 4. Conclusions

In summary, a novel macrolactam (**1**), three dipeptides (**2**–**4**) as well as other alkaloids (**5**–**7**) were isolated from the fermentation liquid of *P. oxalicum* derived from the tuber of *I. trichantha*. Among them, only oxalactam A (**1**) displayed anti-*R. solani* activity at 100 µM with a 29.30% of inhibition rate in vitro (MIC = 10 μg/mL). Notably, the content of **1** is more than 1.70 mg/L in the fermentation liquid of the endophytic fungus *P. oxalicum*. These findings provided an alternative natural resource to obtain novel anti-*R. solani* macrolactam leads.

## Figures and Tables

**Figure 1 molecules-27-08811-f001:**
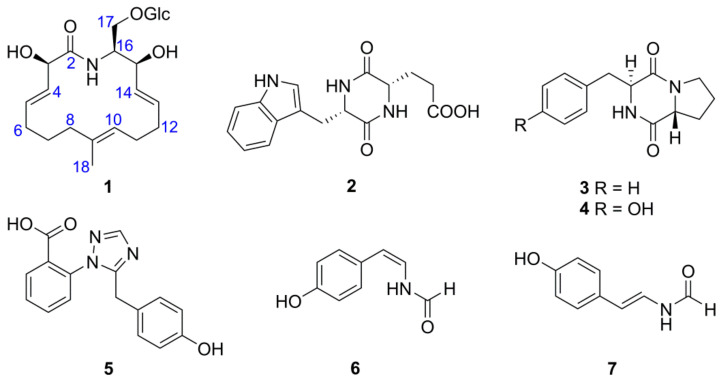
Chemical structures of compounds **1**–**7**.

**Figure 2 molecules-27-08811-f002:**
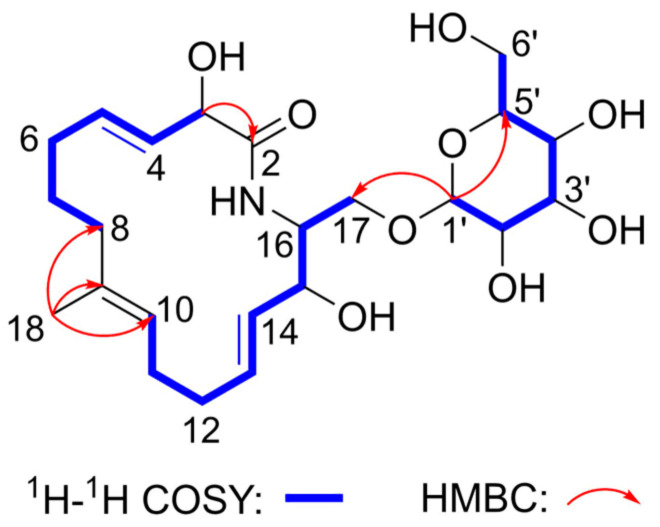
Key ^1^H–^1^H COSY and HMBC correlations of **1**.

**Figure 3 molecules-27-08811-f003:**
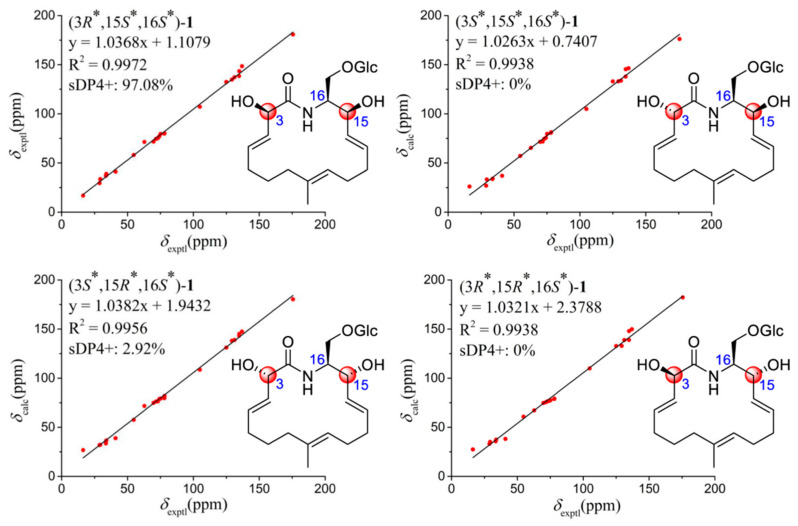
Linear correlation plots of calculated-experimental ^13^C NMR chemical shift values for (3*R**,15*S**,16*S**)–**1**, (3*S**,15*S**,16*S**)–**1**, (3*S**,15*R**,16*S**)–**1**, and (3*R**,15*R**,16*S**)–**1**.

**Figure 4 molecules-27-08811-f004:**
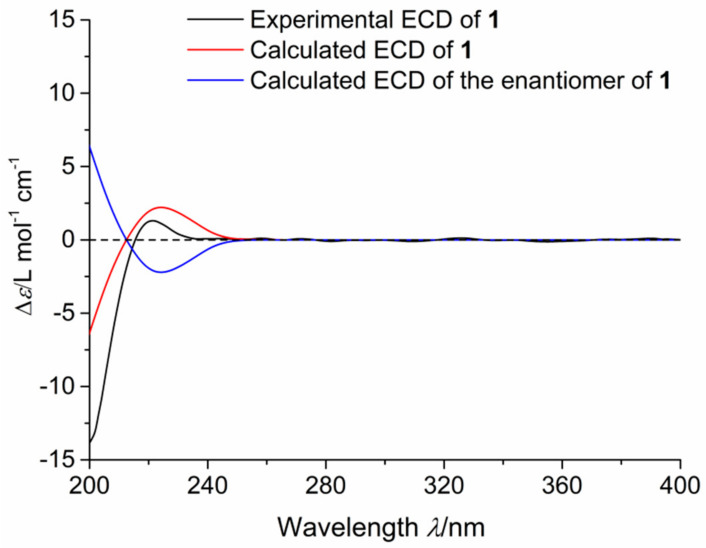
Experimental ECD for **1** and calculated ECD spectra for **1** and its enantiomer.

**Figure 5 molecules-27-08811-f005:**
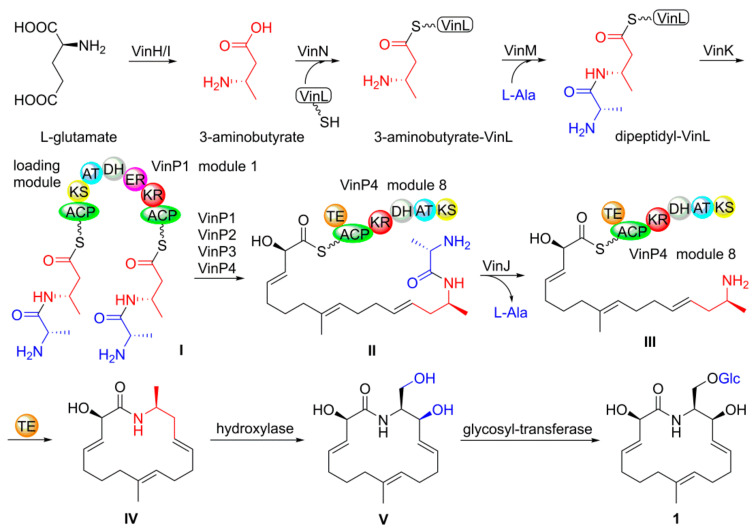
Proposed biosynthesis pathway of **1**.

**Figure 6 molecules-27-08811-f006:**
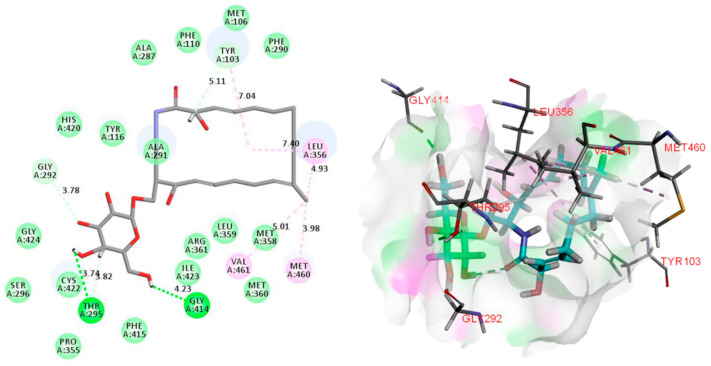
2D and 3D images of oxalactam A (**1**) with the target protein CYP51 (Pink dotted lines: Pi-Alkyl interactions, light green dotted lines: Van Der Waals interactions, deep green dotted lines: Conventional hydrogen bond).

**Figure 7 molecules-27-08811-f007:**
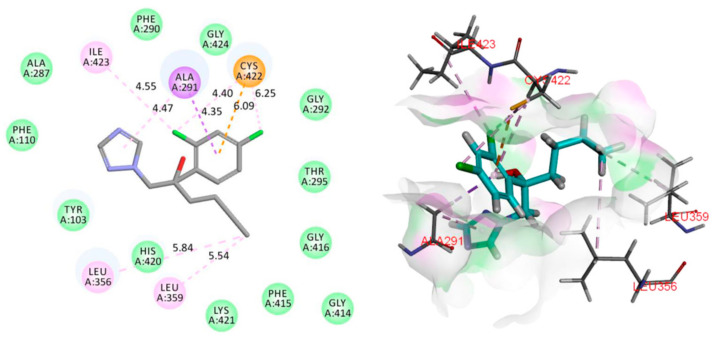
2D and 3D images of hexaconazole with the target protein CYP51 (Orange dotted lines: Pi-Anion interactions, pink dotted lines: Pi-Alkyl interactions, purple dotted lines: Pi-Sigma interactions).

**Figure 8 molecules-27-08811-f008:**
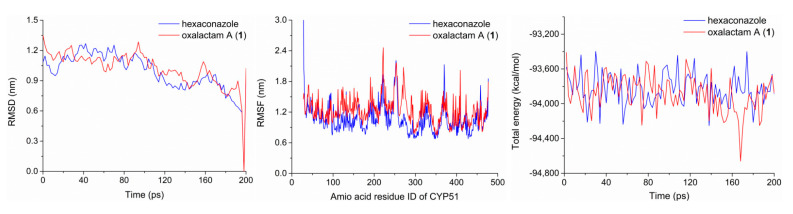
Molecular dynamics trajectory analysis of CYP51–**1** and CYP51–hexaconazole complexes, including the RMSD, RMSF, and the total energy.

**Table 1 molecules-27-08811-t001:** NMR Spectroscopic Data (500 MHz, CD_3_OD) for Oxalactam A (**1**).

Oxalactam A (1)		
Position	*δ*_H_ (*J* in Hz)	*δ*_C_, Type
2		175.6, C
3	4.44 (d, *J* = 6.0 Hz)	74.2, CH
4	5.49 (dd, *J* = 16.0, 6.0 Hz)	129.2, CH
5	5.84 (dt, *J* = 16.0, 6.0 Hz)	134.8, CH
6	2.04, m	33.6, CH_2_
7	1.40, m	29.3, CH_2_
8	1.98 (t, *J* = 7.5 Hz)	41.0, CH_2_
9		136.9, C
10	5.15 (t, *J* = 6.5 Hz)	125.0, CH
11	2.06, m	28.8, CH_2_
12	2.04, m	33.9, CH_2_
13	5.72 (dt, *J* = 15.1, 6.1 Hz)	134.7, CH
14	5.45 (dd, *J* = 15.1, 5.4 Hz)	131.2, CH
15	4.14 (dd, *J* = 10.1, 5.4 Hz)	73.0, CH
16	3.97, m	54.7, CH
17a	3.71 (dd, *J* = 10.3, 3.4 Hz);	69.8, CH_2_
17b	4.13, overlap	
18	1.60, s	16.3, CH_3_
1′	4.27 (d, *J* = 7.8 Hz)	104.9, CH
2′	3.19, m	75.1, CH
3′	3.28, m	78.1, CH
4′	3.27, m	71.7, CH
5′	3.36, m	78.0, CH
6a’	3.86, (d, *J* = 11.7 Hz)	62.8, CH_2_
6b’	3.66, (dd, *J* = 11.7, 4.3 Hz)	

**Table 2 molecules-27-08811-t002:** Anti-*Rhizoctonia solani* activity of compounds **1**–**7** in Vitro *.

No.	Inhibition Rate (%) **	MIC (μg/mL)	ED_50_ (µM)
**1**	29.30 ± 2.91	10	/
**2**	−7.82 ± 1.31	/	/
**3**	2.13 ± 2.29	/	/
**4**	0.83 ± 2.12	/	/
**5**	−3.34 ± 2.12	/	/
**6**	0.83 ± 2.12	/	/
**7**	2.11 ± 2.94	/	/
Carbendazim ***	82.39 ± 7.32	/	/
Hexaconazole ***	70.55 ± 5.8 [33]	/	2.44 [33]

* All measurements were carried out in triplicate. ** The test concentrations of compounds **1**–**7** and carbendazim are all 100 μM and hexaconazole is 10 μM. *** Standard anti-*Rhizoctonia solani* positive control substance.

**Table 3 molecules-27-08811-t003:** The interaction analysis of the molecular docking study on 14α-demethylase CYP51 with oxalactam A (**1**) and hexaconazole.

Compound	Binding Energy (kcal/mol)	Interaction with Amino Acids
**1**	−202.0196	TYR103, THR295, LEU356, GLY414, MET460, VAL461
Hexaconazole *	−105.4279	ALA291, LEU356, LEU359, CYS422, ILE423

* Standard anti-*Rhizoctonia solani* positive control substance.

## Data Availability

Not applicable.

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
