# Peer review of "Oxalactam A, a Novel Macrolactam with Potent Anti-Rhizoctonia solani Activity from the Endophytic Fungus Penicillium oxalicum"

_molecules, 2022, doi:10.3390/molecules27248811_

Round 1
Reviewer 1 Report
The authors reported a novel macrolactam and its antifungal activity. The manuscript is interesting. Detailed comments:
1) The mass assignment [M-H + HCOOH + H]+ is not correct (lose a proton, obtain a proton and add a neutral acid), please check.
2) The experimental CD signal is very weak. So the absolute configuration assignment is not solid. I suggest the author use an another method to confirm the configuration.
3) The compound yields in the main text and SI are different, please keep consistent.
4) For the 16s-rRNA identification, the sequence need to be placed in the SI.
Reviewer 2 Report
There are some issues should be addressed:
1. Line 60: “the tuber of Icacina trichantha” should change to “the tuber of I. trichantha” (the second appearance)
2. Line 61: “Fungi imperficti” is not a scientific name of a species, thus the authors don’t need to Italic in here. And please check again “Fungi imperficti” or “Fungi imperfecti”.
3. Line 62: The authors please check again the genus of “Colletotricnum gloeosporioides” (Colletotricnum or Colletotrichum)?
4. Please discuss the peak at m/z 606.1994 (Figure S10, HR-ESI-MS spectra) in the revised 2.1 section. Why this peak is higher than the peak at m/z 519.2236 [M-H + HCOOH + H]+? And what does m/z 606.1994 means?
5. Please add the meaning of the interactions color in the legend of Figure 6 and 7.
6. Lines 206-208, “the equilibrium state at 0.6-0.9 nm” represents only the information in RMSD plot. The authors please add the information that RMSF plot provides in the molecular dynamics discussion part.
7. The reference of protein PDB ID: 3GW9 is required. The authors should cite this reference in the revised manuscript.
8. Hexaconazole was used throughout your research (positive control in anti-Rhizoctonia solani activity, molecular docking, and molecular dynamics simulation), so what was the role of Carbendazim in your study? Why the positive control was chosen with two compounds?
9. Please add the information of MD software in the section 3.10.
10. Section 3.4, lines 265-266: Compound 1 was enzymatic hydrolysis to identify the configuration of sugar moiety, and the aglycone was purified by HPLC. Thus, please add the NMR data of the aglycone in the main text, and the 1H and 13C-NMR spectrum of the aglycone in the revised version of supplemental material.
Round 2
Reviewer 2 Report
The authors have made a good effort to address all comments by the Reviewer.
However, there is still a major problem in your revised manuscript:
1. In the “Instructions for Authors” of Molecules
(https://www.mdpi.com/journal/molecules/instructions)
“Manuscript preparation
Physical and Spectroscopic Data: Physical and spectroscopic data as well as tables for NMR data should be prepared according to the ACS's Preparation and Submission of Manuscripts standard (page 4).”
The authors can follow this link: https://publish.acs.org/publish/author_guidelines?coden=jnprdf
In the ACS and Molecules standard: “The structures of compounds are expected to be supported by high-resolution mass spectrometry (error limit 5 ppm or 0.003 m/z units)”.
In the revised manuscript (lines 78-79), the authors provide the MS data of compound 1: “the HR-ESI-MS at m/z 518.3077 [M + HCOOH + H]+ (Calcd. for C24H40NO11, 518.2601)”.
The error in HR-MS of compound 1 is 0.0476 m/z. It is more higher than the acceptable error (~ 16 fold).
Thus, the authors must recheck the HR-MS of new compound 1 following the journal standard to confirm the chemical structure of compound 1 is correct.
(Recommend: the authors should carefully calibrate the HR-MS instruments for optimal performance and accurate identification before conducting your sample)
